# Enabling Immersive Exercise Activities for Older Adults: A Comparison of Virtual Reality Exergames and Traditional Video Exercises

Lucie Kruse [1,*], Sukran Karaosmanoglu [1], Sebastian Rings [1], Benedikt Ellinger [2] and Frank Steinicke [1]

1    Department of Informatics, University of Hamburg, 22527 Hamburg, Germany; sukran.karaosmanoglu@uni-hamburg.de (S.K.); sebastian.rings@uni-hamburg.de (S.R.); frank.steinicke@uni-hamburg.de (F.S.)
2    Statistisches Bundesamt—IT Kompetenzzentrum Auswertung und Analyse, 65189 Wiesbaden, Germany; benedikt.ellinger@destatis.de
*    Correspondence: lucie.kruse@uni-hamburg.de

**Abstract:** Participating in cognitive and physical activities can help older adults to live a healthy and independent life. However, with the ongoing pandemic, face-to-face training options became unavailable or limited, yielding a need for alternatives. In this paper, we conducted a user study with older adults ($N = 25$) to compare a traditional, recorded 2D gymnastics video with an immersive virtual reality (VR) exergame. We evaluated the movement and heart rate of the participants, as well as their enjoyment, attention to the task, and perceived workload. In the VR condition, we additionally assessed their feeling of cybersickness. Finally, qualitative feedback about their preferences was collected. The results indicate that our immersive VR exergame can be a suitable alternative, but not a replacement for traditional 2D video-based exercise activities. Furthermore, the cognitive aspect of exergames can lead to the feeling of physical workload, even if easy movements are performed. Finally, we discuss the implications of our results for future VR exergames and point out advantages and disadvantages of the systems.

**Keywords:** virtual reality; exergame; older adults; video exercise

## 1. Introduction

For humans, in particular older adults, it is important to remain fit as long as possible in order to keep autonomy and to be able to live a healthy and independent life. Research has shown that regular exercises can maintain and improve physical and cognitive abilities even for older adults [1–3]. In particular, the combination of physical and cognitive exercises can reduce neurological degeneration and prevent diseases such as dementia [4–6].

During the COVID-19 pandemic, physical training groups, sport teams, access to gyms, and individual physiotherapist sessions were limited, and, therefore, physical activities of older adults have decreased [7]. The unavailability of group-based physical activity, as well as several lock-downs, led to the need for alternatives such as home-based training.

Older adults tend to prefer face-to-face interaction, and the direct support of physiotherapists for exercises [8,9]; however, in times when this contact is not possible, alternatives have to be provided. Exergames, a term used to combine the words "exercise" and "games" [10], are one possibility for home workouts. Younger people are already familiar with online video classes or virtual reality (VR) exergames for training, and 22% of people have used more online fitness videos during the pandemic than compared to before [11]. While there are a lot of exercise videos for older adults online, VR exergames are rarely used by this age group today [9]. Furthermore, 2D video-based training has limitations regarding the possibilities of showing complex movements from different (also egocentric) perspectives or of tracking and recording these motions. Moreover, VR has the potential to provide more engaging and immersive experiences, whereas it also comes with the cost of

more intrusive and bulky technology. Hence, it remains open whether or not immersive VR exergames can provide an acceptable alternative for 2D exercise videos for older adults.

In this paper, we compare a traditional, recorded 2D exercise video to a VR exergame that aims to train similar muscles and body parts. The video shows familiar movements performed by a real trainer, while the VR exergame provides the ability to train 3D movements in an immersive virtual environment (VE), with virtual avatars in front of the user. The goal of our evaluation is to analyse whether VR exergames are an acceptable alternative.

We conducted a within-participants user study with 25 older adults that performed both exercise conditions, and then commented on their perceived workload, attention, and enjoyment, as well as which program they preferred. Furthermore, we analysed the movements and heart rate values measured during both conditions and discussed the advantages and disadvantages of both programs.

Our research was driven by the following research question:

*RQ*: Can a VR exergame be an alternative to a traditional 2D gymnastics video for older adults?

Following our research question, we state the following hypotheses:

**Hypothesis 1 (H1).** *The video and VR exergame will show similar values for perceived workload, attention, and enjoyment.*

**Hypothesis 2 (H2).** *The VR exergame will be preferred by the same amount of people as the traditional video of well-known exercises.*

**Hypothesis 3 (H3).** *Older adults will prefer doing the exercise video for training in long-term use.*

Hypothesis 1 was developed due to the similar length and difficulty of the exercises in the video and the game, which was determined in focus groups and a pre-study. Working with the target users in a human-centred design approach resulted in a VR exergame that shows a high usability for this user group, which we suppose is comparable with traditional exercise videos for this age group.

Regarding hypothesis Hypothesis 2, we suppose that the novelty effects of the VR application will be balanced out by a familiarity with the traditional exercises provided in the video. The personal decision of preference will depend on the participants, their physical fitness, and their openness to new technology. Since our participant group will be taken from the general population of a senior living facility, we expect these factors to be balanced out.

Hypothesis 3 was established because the video exercises are more familiar and easier to access. While most older adults are familiar with exercise videos and use a computer in their free time, VR systems are still an unfamiliar concept, and most head-mounted displays come with usability and technical issues that prevent the older generation from using them without help (e.g., setting up the hardware and starting applications).

To summarize, it is still unclear how older adults will compare a traditional exercise video to an immersive VR exergame, but the latter provides several advantages which will be discussed in the scope of this paper.

The remainder of the paper is structured as follows: first, we will describe important previous work that has been conducted in this field of research. Then, we will describe the hardware and software used in the study, as well as the game and video design. This is followed by a description of the study we conducted. Then, the study results will be evaluated and discussed. Finally, we will talk about the conclusion we drew from this research study and reflect on its contribution.

## 2. Related Work

In this section, we present related work on fitness and serious games for older adults.

### 2.1. Fitness in Old Age

When ageing, many physiological changes occur in the body, which can affect tissues, bones, and organs [12,13]. This process can restrict autonomy, impact activities of daily living, and lead to a lower quality of life [12]. Regular exercise can improve physical and cognitive well-being of older adults [1,2], as well as psychological well-being [12], leading to a more independent and healthy life and reducing the risk of falls or chronic illnesses. Studies also indicate a decreased risk for [14] and reduced progression of [15] dementia when doing physical activities, highlighting their importance for cognitive abilities. *Maestro Game VR* was developed to slow down this progression or help to prevent it, providing both physical and cognitive training.

Group workouts, personal trainers, or going to the gym are traditional training methods employed in senior living homes, but were largely cancelled during the COVID-19 pandemic [16], yielding the need for other programs, e.g., home-based. Home-based exercise studies indicate that these can have psychological benefits [17], help to prevent falls [18,19], and have an overall influence on physical abilities [18,20]. Home-based exercises have been implemented with videotapes [20]; exercise programs [19], e.g., the Otago Exercise Program [21]; or technological applications [22]. One big issue is program adherence [8], with relatively large dropout rates and lower participation (e.g., only 63 % adherence in [19] and 68 % of participants who completed the program at least once a week in [1]). Simek et al. [23] found that program adherence was influenced by the perceived effects of the program on physical and mental health and program structure. Furthermore, personal preferences play an important role [23,24]. The last point is especially important for us, since we aim to provide an alternative exercise method. Some people might prefer it over traditional solutions or would like to change their workout temporarily, therefore contributing to their adherence and motivation.

### 2.2. Serious Games for Older Adults

Serious games have the advantage of motivating the participants to train their cognitive and physical abilities [24,25]. The combination of physio-cognitive exercises according to the dual-task paradigm claims to be an especially effective approach [26,27]. Several exergames have been developed to improve physical and cognitive abilities of older adults [28], people with dementia [29–31] or older adults with Parkinson's disease [32].

In the literature, research has mainly focused on two types of exergames: (1) non-immersive 2D games that use a display or TV and track movements with external devices such as the MS Kinect, and (2) immersive 3D VR exergames.

Gerling et al. [33,34] evaluated the feasibility of an exergame using the Nintendo Wii Balance Board, where the users had to carry out different tasks by shifting their body weight on the Balance Board, and found that this setup was possible. Van Diest et al. [35] examined the feasibility of unsupervised exergame training at home. They used a 2D game displayed on a TV and an MS Kinect for tracking the movements of the participants. The players had to lean to the sides to control an ice-skater on a frozen canal. The authors found that some participants' balance improved during the 6-week study. Unbehaun et al. [36,37,38] evaluated four balance games, where people with dementia performed lower and upper limb movements from the Otago exercise program using MS Kinect. The movements were linked to activities such as walking through a virtual park, steering an air plane, or picking apples and led to improved gait, coordination, balance, and mobility.

VR games provide improved immersion, flow, and motivation compared to 2D displays, especially in younger adults [39,40]. The immersion and 3D representation features of VR can lead to better performance [40]. In a 4-week study, Huang [41] investigated the effect of immersive virtual environments (IVEs) on various cognitive outcomes in older adults. Their findings showed that exergame training in IVEs can improve performance

in inhibition and task switching compared to a non-immersive version, and the feeling of presence can be a mediator of this outcome. VR exergames can also be used for people with dementia because of their promising results in improving cognitive and physical skills. For example, on a VR farm, participants in the work of Eisapour et al. [24,42] carried out reaching and rowing movements to perform activities such as sorting fruit into boxes or rowing a boat. Physiotherapists commented positively on the participants' range of motion. Similarly, Memory Journalist VR [43] is an exergame where the players need to take photographs of well-known sights in VR. It was developed for people with dementia [44], and in a 9-week study, the authors showed that cognitive as well as psychological well-being of the players improved more than those of a control group that did not play the game [29].

While VR technology is promising to use in this context, it also comes with usability and cybersickness problems [45]. Although these problems might be partially avoided [46], and in particular usability problems might be easy for younger people to solve, they can be challenging for older adults due to ageing and having little experience with interacting with innovative technology [47]. Therefore, in our research, considering the advantages and disadvantages of both video and VR exergame based exercises, we aim to compare these home-based exercise options from the perspective of older adults.

## 3. Materials and Methods

In this section, we will describe both the VR exergame and the exercise video, as well as the user study.

### 3.1. VR Exergame: Maestro Game VR

*Maestro Game VR* was developed with a human-centred design approach, with several focus groups, interviews with older people and experts, and prototyping sessions of previous versions of this game, as described in our previous paper [25]. Originally thought as a game for people with dementia, we had to slightly change it for people without dementia, since the COVID pandemic prevented us from testing it with this user group. Our focus now is more on preventing physical and cognitive diseases through movement. We therefore changed the difficulty, removed some help options to facilitate the start of the game and sped up the game flow.

The exergame was implemented using C# in Unity (version 2019.4)[1] with the High Definition Rendering Pipeline and Visual Effects Graph with Particle Strips. The exergame was played using an i9-9900K CPU and a GeForce RTX2080 Ti graphics card. We used the Valve Index VR headset, two controllers to act as batons, and two base stations. The player's in-game view was displayed on a 65-inch 4K display for the spectators.

In *Maestro Game VR*, the players are located in a 3D concert hall. They are playing the role of a conductor that has to conduct a band of three musicians by following a virtual 3D path in front of them with a baton. The virtual path is made up of three essential parts, as can be seen in Figure 1:

- A disco ball which the player is supposed to follow with the tip of the baton.
- A tunnel with staff lines extending before and after the disco ball to help the player anticipate the target's path.
- Music notes that regularly spawn along the tunnel to calculate the player's accuracy when the disco ball hits them.

In order to calibrate a fitting path for the player, a calibration is performed at the start of the game, where players have to move their arms as far to the front, sides, and up as possible. With these measurements, the shape and position of the path is determined. For each arm, an ellipsoid with the shoulder position as the centre and the maximum arm reach as outer points is created (see Figure 2, light green). A cuboid shape that begins a little in front of the shoulder position and spans to the outer edge of the extended arm is then added (red cube). The length towards the other side and in the downward direction can be configured with values between 0 (ends at the shoulder position) and 1 (extends as far to the other side as possible), to enable different difficulties. All points within the

intersection of the light green circle and the red cube are possible points the Bézier curve can go through.

In accordance with physiotherapists, useful but comfortable movements with occasional more difficult spikes are procedurally generated. This approach does not put too much strain on the player's muscles, but also includes some challenges with the prospect of improvements.

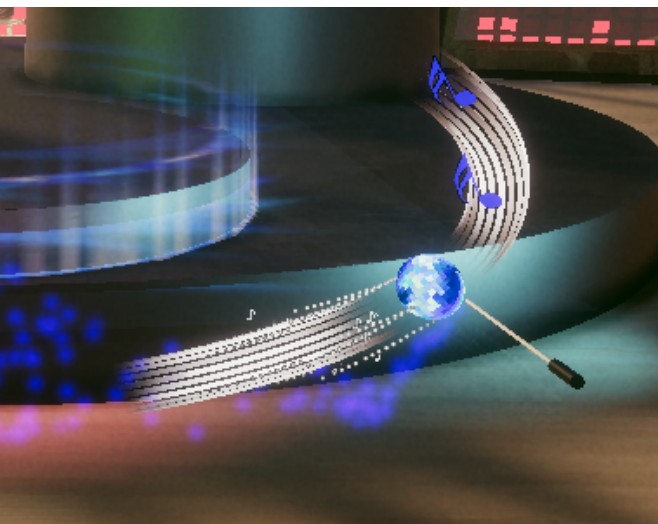

**Figure 1.** The disco ball that the player has to follow on a virtual path that resembles staff lines. The baton emits blue sparkles to highlight that the right hand has to be used. Notes are used as measurement points for accuracy.

The players are only conducting with one hand (left or right) at a time, but change hands after 30 s of successfully conducting the musicians. This way, they can fully focus on following the path correctly, reducing the physical and cognitive complexity and their frustration. Oral instructions are given for a hand change, and the colour of the disco ball and baton sparkles changes (blue for the right hand and red for the left hand).

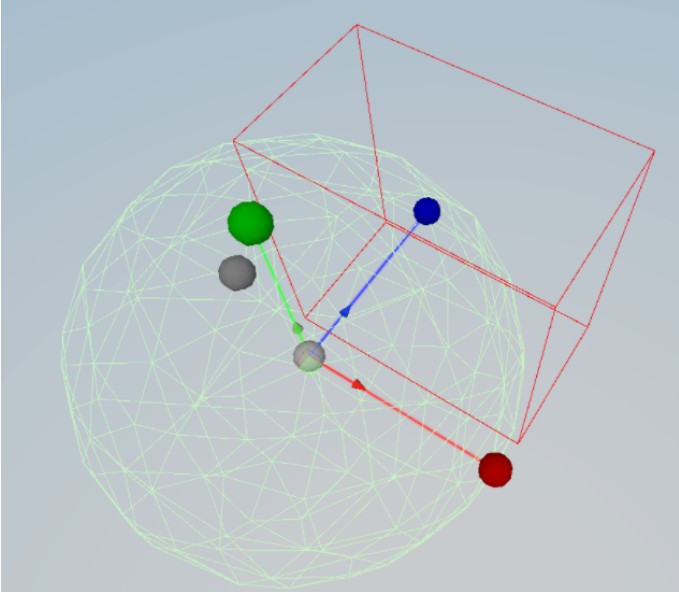

**Figure 2.** Calibration and curve creation. Light grey: shoulder position of the player; blue: forward; green: upward; red: right side. Light green lines represent the maximum reach of the player's arm, and the red cuboid shape shows the points that can be chosen for the current configuration. With these, Bézier curves are procedurally created.

The usage of music has been implemented in several other exergames, such as Beat-Saber[2] or OhShape[3], and has been shown to increase motivation and performance and invoke old memories [31,32]. The players can choose one out of eleven songs, which are a variety of newer and older pop, classic, and rock songs. In prototyping sessions and interviews, we asked the participants for their favourite music and tried to include it in the game. The songs are split into the individual music instruments using *Spleeter* [48], so that they can be used individually for the three musicians. Depending on the instrument, the beat changes, resulting in different movement rhythms of the disco ball.

The active musician is always located at the front of the stage, with the two other musicians behind them (see Figure 3). When the player successfully follows the disco ball, the volume of this musician becomes louder depending on the distance of the baton to the centre of the disco ball. When the maximum volume is reached, the stage starts to rotate and the next musician moves to the front. When the musicians are not actively conducted, their volume decreases again. The goal is to increase the volume of all musicians to its maximum, resulting in the "full" version of the song, which can then be heard as a reward. If the player stops following the path or if the baton deviates too far from the disco ball, the musician stops and a distorted sound of their instrument is played. When a song is completed until the end, confetti starts to fall from the ceiling, fire appears at the sides of the stage, and a long applause is played. A new song can then be chosen.

The player's movements, personal settings, and their calibration are tracked, enabling a later evaluation and the possibility to compare their progress.

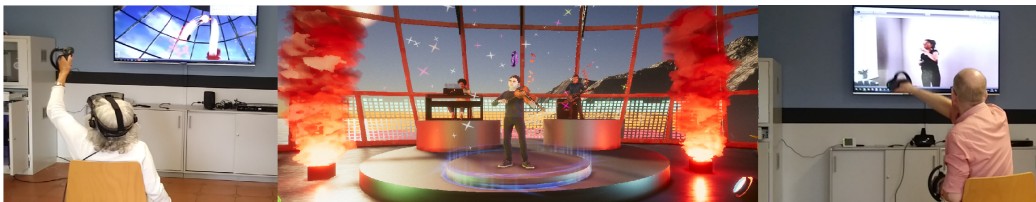

**Figure 3.** Left: An older adult playing *Maestro Game VR*. Middle: The virtual stage. Right: An older adult doing his workout with the 2D video.

*3.2. Exercise Video*

We chose to compare the VR exergame with a 2D gymnastics video because it creates external validity for our research, as many people do their workouts with exercise videos due to the COVID-19 pandemic [11,49]. The exercise video was recorded using an iPhone 7 Plus. It was displayed on a 65-inch 4K display, and two Valve controllers were used for tracking the hand movements of the participant. The video featured exercises adapted from different sources [21,50–52]. In detail, the exercises and number of repetitions were:

- Side arm raises (5×);
- Neck lateral flexion stretch (5×);
- Neck rotation to the sides (5×);
- Neck flexion/extension forward and backward (5×);
- Shoulder circles to the front (7×);
- Shoulder circles to the front, including elbows (7×);
- Shoulder circles to the back (7×);
- Shoulder circles to the back including elbows (7×);
- Arm circles to the front (6×);
- Arm circles to the back (6×);
- Torso rotations (7×);
- Punches to the upper sides (5×);
- Overhead punches (7×);
- Side arm raises (5×).

To replicate a realistic as well as physically benefiting experience, the video and the movements were inspired by several YouTube videos [51–54]. The senior living home employees, including physiotherapists and caregivers, reviewed and agreed to the video and exercises.

The video was integrated into a Unity application to enable tracking, data collection, and individual music selection. Even though the video was pre-recorded, the participants were able to freely choose three out of 12 possible songs of different genres as background music. The program automatically tuned down the volume of the songs while oral instructions were presented, so they were clearly audible.

### 3.3. User Study

The goal of this user study was to compare a traditionally recorded 2D gymnastics video with a VR exergame regarding enjoyment, attention allocation, perceived workload, and preference. All participants were older adults still living independently in their own apartments, which are part of a local senior living facility that offers serviced homes. Our project works with various ethics and privacy experts, as well as the local senior living facility, who support us during the project. We contacted our local ethics department and received an acknowledgement for the study. Furthermore, an ethics workshop was held before the start of the study to discuss ethical as well as security and privacy concerns for all of our studies, and advice from this workshop was integrated.

#### 3.3.1. Pre-Study

Four people ($P_1$–$P_4$, all female, average age: $81.75 \pm 4.19$ years) participated in a pre-study to determine a suitable length and difficulty for the game. We planned to play the game in a standing position, but two of the (first-time VR) users asked to sit down after putting on the head-mounted display (HMD) because they felt insecure, and sitting exergames can also provide a high usefulness to players [55], so we chose this setup for all following players.

Furthermore, we originally planned to play three songs in a row, as it was suggested by our players in the prototyping sessions. However, the players in the pre-study showed signs of exhaustion and cybersickness during the second and third song (increased breathing rate, increased heart rate, and dizziness) and also reported those feelings after the study. This is also consistent with literature, which indicates higher cybersickness scores for inexperienced VR users [46]. One more experienced user said that three songs were suitable for her, but since we included experienced and less experienced VR users, and older adults with varying physical fitness, we decided to choose a safe setup and only play one song. After all, this study was not meant to test efficiency of the game regarding physical training effects, but to compare the feasibility and preference of a traditional 2D gymnastics video to a VR exergame. As determined in prior prototyping sessions and in the pre-study, we also adjusted the speed and difficulty of the game to make it safe and enjoyable for all older adults, regardless of physical fitness.

#### 3.3.2. Participants

In the actual study, 25 older adults took part ($P_5$–$P_{29}$, 3 male, 22 female, average age: $81.24 \pm 4.97$ years). Five older adults already had experience with VR and participated in some of our prior studies and prototyping sessions. Three participants reported never doing any sports, fifteen were doing sports at least once a day, and seven exercised at least once a week. Activities mainly included yoga, gymnastics, walking, or biking. Regarding gaming, seven people said that they did not play any games (computer or board games), four people played every day, twelve people at least once a week, and two people at least once a month.

### 3.3.3. Measurements

Several measurements were used to compare the two conditions. To evaluate the well-being of the participants after playing the VR-exergame, they filled out the Simulator Sickness Questionnaire (SSQ) [56] before and after the VR experience. In order to assess their enjoyment, the participant answered three questions on a scale from 1 to 5 ("The task was fun", "The task was interesting", "The task was entertaining"), adapted from the interest/enjoyment sub-scale of the Intrinsic Motivation Inventory (IMI) [57,58]. Furthermore, the sub-scale Attention Allocation from the 4-item scale of the MEC Spatial Presence Questionnaire (MEC-SPQ) was presented: ("I devoted my whole attention to [the medium]", "I concentrated on [the medium]", "[The medium] captured my senses", "I dedicated myself completely to [the medium]") [59]. We chose this questionnaire because of its ability to compare different media, e.g., video and VR, as opposed to well-known presence questionnaires that are mostly used only for VR [60]. Finally, the Nasa-TLX questionnaire [61] was used for an evaluation of the perceived workload.

During both conditions and during a resting period, we measured the heart rate of the participants to have an additional indicator of exhaustion and physical stress, and to better compare the exertion of both programs. This was done using the Polar OH1 heart rate sensor[4], which the participants wore around their arm. Furthermore, we wanted to have a physiological measure to compare to the subjective measurements given in the Nasa-TLX questionnaire.

### 3.3.4. Procedure

The procedure for the study can be seen in Figure 4. We tested the participants in a within-participants design, meaning that each participant tried out both conditions in a counterbalanced way. A consent form was given to the participants a few days in advance so they could read it, sign it, and write down questions. Upon arrival, the older adults filled out a demographics questionnaire and they were allowed to choose one song for the game and three songs for the workout video. Then, we gave them the heart rate sensor and measured their resting heart rate for five minutes. Afterwards, their first condition started.

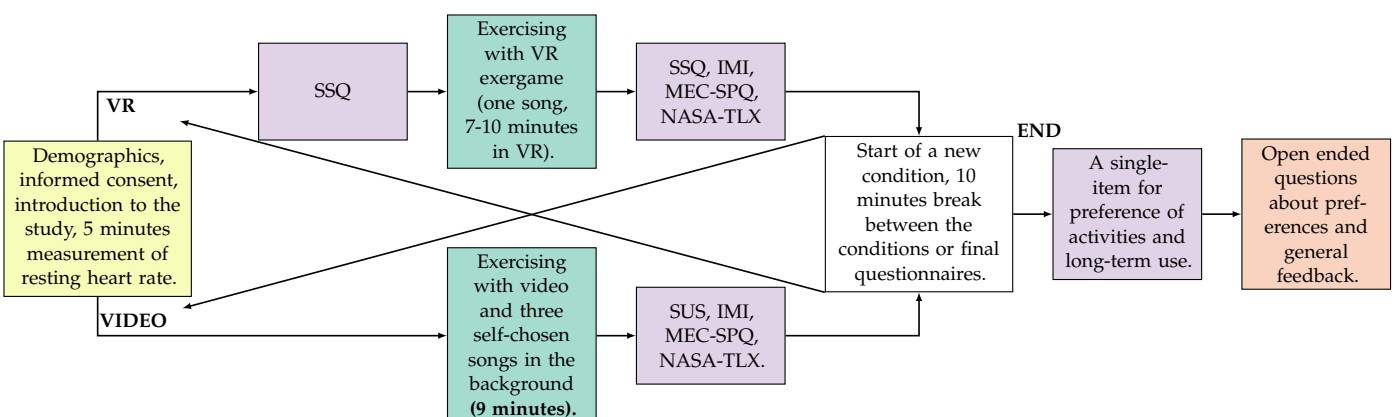

**Figure 4.** Procedure for the study on a comparison of a 2D gymnastics video to a VR exergame.

In case of the exergame, the participants filled out a before measure of the SSQ. Then, they watched a short demonstration video of the experimenter playing the game, so they knew what to expect and to reduce their potential fear. They then sat down on a chair in the middle of the tracking space, put on the HMD, and adjusted it to fit on their head. Afterwards, a calibration sequence was started where a virtual agent asked them to move their arms as far forward, up, and sideways as possible, followed by a tutorial. The tutorial explained how to follow the disco ball with the baton and that the task was to move in a certain changing rhythm. A hand change was also practised. Afterwards, the song they had chosen was started, and they conducted it until the end. When the song finished, they

had some more time to look around in the virtual environment and then took the HMD off again. Then they filled out another SSQ, followed by the interest/enjoyment sub-scale of IMI, the attention allocation part of MEC-SPQ, and the Nasa-TLX. After a 10-minute break, the next condition was started.

In the case of the video condition, the participants again took a seat on the chair in the middle of the room. Two Valve controllers were given to them to track their movements and the video was started. The music that the participant had chosen for the video was played in the background to make the exercises more comparable. After the video, the participants again filled out the IMI, MEC-SPQ, and Nasa-TLX.

After the last condition, they were asked which sports program they preferred and why, which sports program they would prefer to do in long-term use and why, and lastly, to provide additional feedback about the game.

The study took around one hour, with 7–10 min spent in VR, depending on the chosen song length, and a duration of 9 min for the gymnastics video, as suggested in [40].

## 4. Evaluation

In this section, we will report the results of the study. We will report both statistical results from frequentist statistics ($\alpha = 0.05$) and from the Bayesian approach. Frequentist statistics is still the standard in HCI research, but falls short in evaluating in favour of the null hypothesis or depicting how much more likely one condition is over another [62,63]. Evaluation was done using JASP[5]. The descriptive values of the questionnaires can be seen in Table 1.

**Table 1.** The descriptive values of the questionnaire measures and heart rate for both conditions.

| Conditions | Pre-SSQ | | Post-SSQ | | IMI | | MEC-SPQ | | NASA-TLX | | Heart Rate | |
|---|---|---|---|---|---|---|---|---|---|---|---|---|
| | Mean | SD | Mean | SD | Mean | SD | Mean | SD | Mean | SD | Mean | SD |
| VR Exergame | 8.98 | 11.48 | 8.08 | 11.46 | 4.57 | 0.63 | 4.44 | 0.72 | 19.77 | 11.91 | 76.77 | 9.8 |
| Exercise Video | — | — | — | — | 4.6 | 0.68 | 4.63 | 0.56 | 17.23 | 10.94 | 81.97 | 11.56 |

### 4.1. Questionnaires

Both programs, the video and the VR exergame, received similar results regarding enjoyment. No significant differences could be found using frequentist statistics ($t(24) = -0.165$, $p = 0.871$). A Bayesian paired samples *t*-test showed that the null hypothesis (enjoyment$_{game}$ = enjoyment$_{video}$) is 4.68 times more likely than the alternative hypothesis. Using Jeffrey's criterion, this is positive evidence in favour of the null hypothesis [63,64].

For Attention Allocation, there was no significant difference between conditions ($p = 0.122$, $t(24) = -1.605$). A Bayesian *t*-test shows weak evidence in favour of the null hypothesis (attention$_{game}$ = attention$_{video}$), with a Bayes factor of 1.54. The exergame was perceived as cognitively and physically more exhausting than the video. In detail, the video received a Nasa-TLX score of $22 \pm 22.31$ out of 100 points for cognitive exhaustion (median: 10), while the exergame received $27 \pm 21.21$ points (median: 25). For physical exhaustion, the video was rated with $19.6 \pm 20.41$ points (median: 10) and the exergame with $20.8 \pm 20.4$ points (median: 15). Additionally, the perceived frustration was lower during the VR exergame ($11.8 \pm 8.65$, median: 10) than during the video ($15.2 \pm 20.28$, median: 10). While most older adults rated both conditions relatively low, there are some higher ratings of 95 out of 100 points in the cognitive or physical demand. Due to different cognitive states and physical fitness of older adults, these ratings should also be considered for future implementations.

No significant differences between the two conditions could be found ($t(24) = 1.237$, $p = 0.228$). The Bayesian paired samples *t*-test shows a Bayes factor of 2.398 in favour of the null hypothesis (workload$_{game}$ = workload$_{video}$). A positive correlation could be found between physical and cognitive workload in the exergame ($r = 0.832$, $p < 0.001$, $BF_{10} = 65{,}044$), but not for the video ($r = 0.066$, $p = 0.754$, $BF_{10} = 0.26$).

For the exergame, we asked the participants to fill out the SSQ before and after the experience. A non-significant difference ($t(24) = 0.321$, $p = 0.75$) indicates that the game did not cause any cybersickness. A Bayesian $t$-test resulted in a Bayes factor of 4.52 in favour of the null hypothesis ($SSQ_{before} = SSQ_{after}$). We also report the descriptive before (nausea: $M = 6.4872$, $SD = 10.92$; oculomotor: $M = 7.8832$, $SD = 12.47$; disorientation: $M = 9.4656$, $SD = 13.76$) and after values of sub categories of SSQ (nausea: $M = 2.2896$, $SD = 5.7$, $p = 0.04416$; oculomotor: $M = 8.4896$, $SD = 11.64$, $p > 0.05$ ; disorientation: $M = 9.4656$, $SD = 19.69$, $p > 0.05$).

No correlations with age, gaming experience, or the amount of times the older adults exercised were found.

### 4.2. Heart Rate Data

The descriptive values of heart rate measures can be seen in Table 1. It was higher during the video than during the exergame ($t(24) = -4.4$, $p < 0.001$, $BF_{10} = 257.99$ in favour of the alternative hypothesis $HR_{Video} \neq HR_{Game}$), with a mean difference compared to the resting heart rate of $-3.75 \pm 5.22$ during the game and $1.62 \pm 6.76$ during the video. A positive correlation ($r = 0.436$, $p = 0.029$, $BF_{10} = 2.339$) was found between the heart rate during the video and the perceived performance measure from the Nasa-TLX, indicating that a higher heart rate was associated with a better performance. No correlation was found between the heart rate and the perceived physical exhaustion for the video ($r = -0.259$, $p = 0.212$, $BF_{10} = 0.52$) or the exergame ($r = -0.246$, $p = 0.236$, $BF_{10} = 0.482$), or between the other subsections of the Nasa-TLX.

### 4.3. Movement Data

During both programs, the movement of the participants was tracked with the two Valve controllers. The mean distance travelled by the hands of the participants was $187.17 \pm 89.71$ m during the video and $24.22 \pm 4.32$ m during the game, which is a significant difference ($t(24) = 9.219$, $p < 0.001$, $BF_{10} = 5,002,000$). These large differences are mainly due to the longer length of the exercise video, but also the more diverse and bigger movements performed in this condition. Additionally, some tracking inaccuracies due to fast movements occurred. The right hand, which was the dominant hand for most participants, covered more distance than the left one, with $97.78 \pm 46.47$ m for the right hand and $89.36 \pm 46.67$ m for the left hand in the video, and $14.5 \pm 9.71$ m for the right and $9.71 \pm 3.04$ m for the left hand in the exergame condition.

The plots in Figure 5 show the movements of the hands in polar coordinates for both scenarios, the exergame and the video, recorded for $P_{28}$. Since the movement of all participants depended on the individual range of motion, a pooled plot would disguise the results, which is why we will explain the movements of this participant. The other participant's plots show similar results.

The left figure is a histogram that shows how much movement occurred in a specific area. The right figure displays the trajectories of the left and right hand. Additionally the intensity of the movement can be seen. The darker the colour, the more intense the movement. Intensity is measured as a function of movement radius (stretching of the arm) and lifting of the arm. It is considered that lifting or stretching the arm is more intense than flexing the arm. The $0°$ direction is considered to be the centre of the body.

When comparing the video scenario with the game scenario, it can be seen that the video seems to be more intense than the game scenario. This can be explained by the high radius values. However, when looking at the histogram, one sees that the video scenario leads to a narrowed range of motion; the movement takes place in a range of $0°$ and $30°$. In comparison, the game shows a much wider range of motion between $0°$ and $60°$.

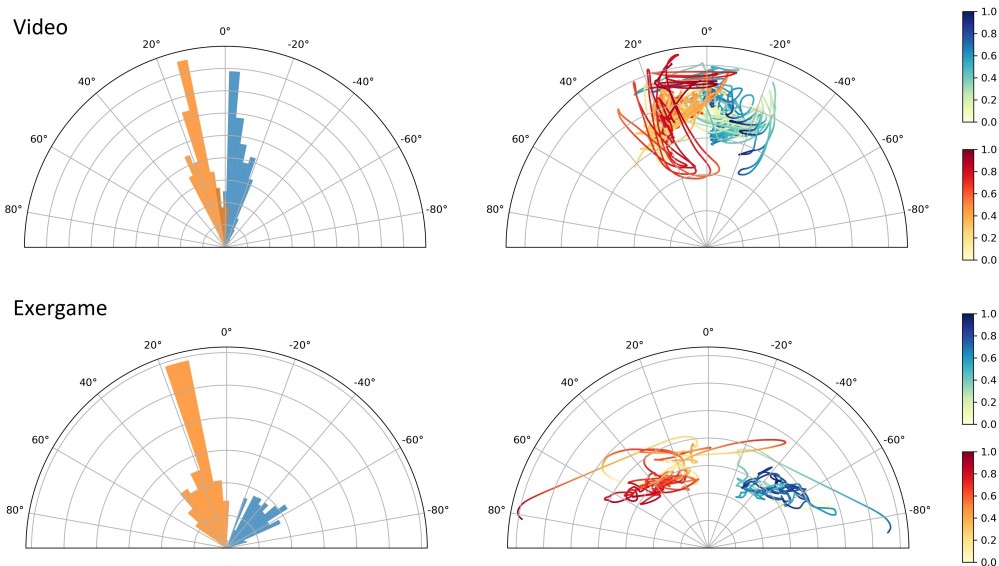

**Figure 5.** Polar plots of the movement of $P_{28}$ for the video (upper two plots) and for the game (lower two plots). The first plot always shows the histogram of the hand movement, while the second one displays the hand movement trajectory with intensity measure.

### 4.4. Qualitative Feedback

On the question of which sports program the participants preferred, 13 people chose the video and 9 people chose the exergame. Three people ticked the boxes of both options, because they could not decide on one. Several people mentioned that they did not understand the question, since they did not perceive the exergame as sports, but just as a game. Those people that preferred the *video* noted that they already knew this kind of exercise ($P_{11}$). Additionally, the exercises trained the whole upper body, as opposed to the exergame where mainly the arms and torso were used ($P_{14}$). It therefore felt more like a workout with a higher physical demand ($P_7$, $P_{17}$, $P_{27}$, $P_{28}$). Additionally, they noted that the exergame needed some getting used to ($P_{15}$), as opposed to the more realistic, face-to-face video ($P_7$, $P_8$). The participants that preferred the *exergame* commented that it was more fun ($P_5$, $P_{21}$) and more interesting ($P_6$). They also commented that compared to the exergame, the video felt more monotonous ($P_9$). It also required more cognitive attention to follow the path correctly ($P_6$, $P_{13}$), which was pointed out positively. Additionally they noted that they liked the virtual environment ($P_9$, $P_{18}$, $P_{21}$), and the music fit well to the movements ($P_{22}$).

For long-term use, 16 participants would prefer the video, and 7 would prefer the game. This time, two people chose both options. The main arguments for the *video* were that the exercises were known ($P_{11}$, $P_{22}$, $P_{23}$), more intense ($P_{16}$, $P_{21}$), and easier ($P_{19}$). Additionally, they showed more variety ($P_{27}$). For the *exergame* in long-term use, the participants again mentioned that it was more fun ($P_9$, $P_{13}$), that they liked the movements ($P_{10}$, $P_{12}$), and that it required more concentration ($P_6$, $P_{13}$). One participant wrote that it was "self-explanatory" why this was the preferred option ($P_{18}$). $P_{22}$, who preferred the exergame, but would use the video for long-term, suggested that the video was more suitable in the long-term usage, but the exergame could be used as a change or for relaxation.

## 5. Discussion

Our goal was to find out if a VR exergame can be an alternative for traditional exercise videos. In particular, during the COVID-19 pandemic, where group and personal sport experiences were limited, many people used exercise videos to remain fit at home.

As expected in Hypothesis 1, almost all results from our qualitative and quantitative evaluation remained very similar, but the results obtained in the subcategory of MEC-SPQ surprised us. VR is known to provide a large feeling of presence [65], which includes turning the user's attention to the medium. The video, on the other hand, also proved to capture the users' attention and even received slightly higher scores than the VR exergame. It seems as though the older adults, who grew up with a TV and mostly use it everyday, maintain the ability to be captivated by it, as opposed to younger people who are already familiar with VR and value its ubiquity. This is similar to what Xu et al. [39] found in their study with young and middle-aged people, where younger players were more immersed in VR than middle-aged ones.

Prior research has shown that VR provides a higher feeling of flow and more enjoyment than a TV display [40]. This effect could not be confirmed in our study, but it has to be noted that our two training applications showed many differences in the task, environment, and movements. Future studies should compare an exercise video to an exergame that also features similar movements.

Hypotheses 1 and 3 could also be confirmed, with around 40 % of the participants preferring the exergame over the exercise video, and around two thirds of participants preferring the exercise video for long-term use. Movement plots support this, with the video showing more intense movements that are more suitable for long-term training. On the other hand, movement precision and range of motion in the video condition are limited due to missing feedback and encouragement for larger movements.

Advantages of the exercise video are that there are many videos available online, which foster different topics and can be watched independently at home. A disadvantage is that there is no intuitive way to control whether the exercises are performed correctly. This could also be seen in our movement analysis, where participants often performed the movements at a different speed or with the reverse arm (see Figure 6a). For this video, these errors did not matter, but for more difficult movements or with specific diseases, the correct execution of the movements is especially important. Additionally, the movements in the video either required oral explanations, or the participants constantly looked at the TV to follow the movements, even if they were supposed to rotate their torso to a different direction (see Figure 6b). For the oral explanations to be audible, the volume of the music had to be decreased, which could potentially break the flow of the experience.

For the VR exergame, movement could be controlled and immediate feedback could be given. If the players did not perform the correct movements, the musicians stopped playing. On the other hand, if the player did well, there was applause and confetti, and the speed of the movements increased. Therefore, the game was able to adjust itself to the abilities of the participant and encouraged them when correct movements were performed. Furthermore, because of the stereoscopic 360° 3D representation of the virtual environment, the participants' ability to replicate spatial movements in their peripersonal space was encouraged. When seeing the required movements in 2D in the video condition, spatial information such as depth of movement on the Z-axis was lost, which is important to create cognitive representations of space [66]. Additionally, prior studies have shown that distance perception in peripersonal space in VR shows smaller errors than in extrapersonal space [67], which, due to the distance of the TV, the exercise video falls into. Additionally, the movements in this game were synchronized with the music, amplifying the feeling of flow and even encouraging some players to tap their feet or to sing along. On the other hand, the movements could become boring because there was not a lot of variety in the used body parts. The initially chosen difficulty was too easy for some participants, but our pre-study showed that the originally planned setup was too difficult for some others. Therefore, the correct starting difficulty and the amount of songs that can be played should be balanced out and decided by the older adults themselves if the game is played outside of the study context. This was also reflected in the heart rate measurements, where the heart rate even decreased compared to the resting heart rate. This might be because many participants were excited to try VR for the first time and did not know what to expect.

However, they started to relax when they were wearing the HMD and when they followed the relatively slow motions of the exergame. Thus, our findings are in line with the prior work [68] that recommends testing VR games with participants before taking baseline physiological measurements and conducting the actual experiment.

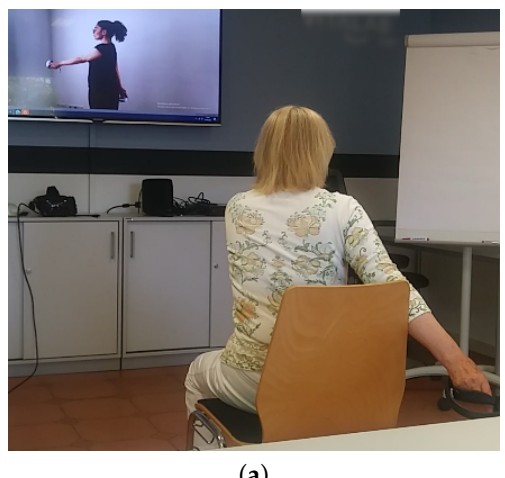 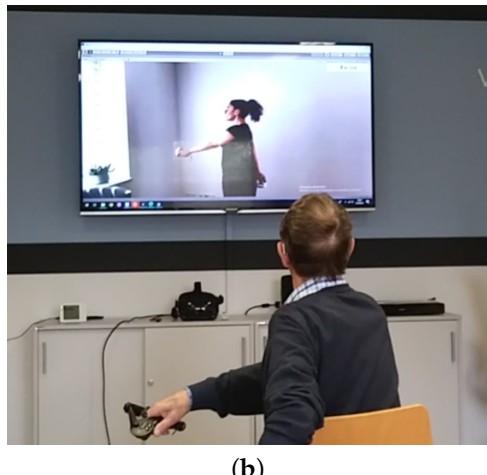

(**a**)                                                                                      (**b**)

**Figure 6.** Two participants doing the exercise video. (**a**) A participant doing the exercises in a reversed way compared to the video. (**b**) A participant looking forwards while doing torso rotations.

Interestingly, our findings point out that while older adults rated the VR exergame and video based exercise similar in terms of perceived physical workload, the heart rate and movement values for the video-based exercise were significantly higher than for the VR exergame. We speculate that the similar rating of perceived physical workload for the VR game might be due to the unfamiliarity with the given VR exergame mechanics and VR technology, although long-term studies are needed to verify this claim. Contrarily, prior research [39] was not able to find an influence of display type on perceived workload. Furthermore, the cognitive workload during the game might have influenced their perception of the physical workload, even though this is also not supported by the literature [69]. Both cited studies were tested with younger adults, yielding the need for studies with an older user group. We conclude that the novelty effects of VR with older adults should be considered in future work, as it can be overwhelming or exciting and can thus affect the overall experience.

To discuss safety issues, it is important to follow the guidelines suggested by the HMD manufacturers. Additionally, the frailty of this user group has to be considered. Choosing a standing setup should therefore only be proposed to players that feel secure and have used VR before. This should always be their own decision, with options to play the exergames in sitting and standing positions, and to switch between the two whenever necessary. Future exergames should adapt to this dynamically. When employed at a senior living home, the tracking space should be secured, e.g., with the help and involvement of technicians [44].

Our findings suggest that the VR exergame is not able to fully replace the traditional workout. However, with the demographic change, caregivers need to be relieved, and additional offers should be available for independent training of the older adults. Our study showed that virtual humans or virtual content was largely accepted by the older adults, which is also supported by the literature [44,70]. With new generations, this trend of accepting new technologies is likely to continue. In combination with the dynamically changing difficulty, and the ability to receive immediate feedback about movements and track the older adults' progress, VR exergames are shown to be a suitable alternative.

*Limitations*

The exercises performed in the video are not exactly comparable with those in the VR exergame. While the exergame mainly focused on arm movements and some torso rotations, the video provided more diverse exercises with significantly more movement. Yet, we note that the exercise movements in the video represent a real-world-based scenario, and the game could be adjusted to feature more and larger movements. Additionally, the question of which *sports program* the participants preferred resulted in a surprise, because the VR exergame was not perceived as sports, but rather as a game, as mentioned by several participants. This might have distorted the results of this question towards the participants choosing the video. Another point which was apparent in the questionnaire results was a ceiling effect. Both programs received very high scores, with many questions receiving the maximum amount of points. So even if the second program the participants experienced was in their eyes "better", there was no way to correct their prior evaluation of the first program. Lastly, we only had three male participants in our study, so gender differences cannot be accounted for. On the other hand, we would like to emphasize that because of the longer life expectancy of females, they are the main population of senior living homes and therefore more likely to use the systems.

## 6. Conclusions

We have developed and evaluated a VR exergame that features rhythmic movements in 3D space and compared this to a traditional 2D gymnastics video. Both programs received similar scores regarding enjoyment, workload, and attention, and neither a significant difference using frequentist statistics nor a high likelihood of differences using Bayesian statistics could be found. While the video was preferred by slightly more people (13), the game also received nine votes for preference (three people remained undecided). For long-term use, more people (16 vs. 7) preferred the exercise video.

In a qualitative evaluation, the participants commented positively on the fun and novelty of the VR exergame, while some also found the VR exercises too repetitive. Regarding the exercise video, the older adults appreciated the known exercises and their variety, as well as the higher physical demand.

The different nature and exertion of the exercises in both programs is one of the limitations in this study. Furthermore, even though the perceived physical workload in both programs was the same, some participants did not perceive the game as exercising. Future studies should develop and evaluate VR games that feature similar movements as traditional workout videos. Finally, long-term studies should be conducted to test VR and video exercises without the potentially confounding novelty effect.

In summary, our results showed that although exercising with VR exergames, or using VR technology in general, is not a familiar concept for older adults, VR exergames are promising and can be an alternative option to exercise for this user group. We recommend using VR exergames for people who like to experience new emerging technologies or alternatives and need or want an adjustable, dynamic fitness application.

**Author Contributions:** Conceptualization, L.K., S.K., S.R. and F.S.; Data curation, L.K., S.K. and S.R.; Formal analysis, L.K., S.K., S.R. and B.E.; Funding acquisition, F.S.; Investigation, L.K., S.K. and S.R.; Methodology, L.K., S.K. and S.R.; Project administration, S.R. and F.S.; Resources, L.K., S.K. and S.R.; Software, L.K., S.K. and S.R.; Supervision, F.S.; Validation, L.K., S.K., S.R., B.E. and F.S.; Visualization, L.K.; Writing–original draft, L.K.; Writing–review & editing, L.K., S.K., S.R., B.E. and F.S. All authors have read and agreed to the published version of the manuscript.

**Funding:** This project was supported by the German Federal Ministry of Education and Research (BMBF) as well as the German Research Foundation (DFG), the European Union's Horizon 2020 research and innovation program and the Federal Ministry for Economic Affairs and Energy (BMWi). The author acknowledges the support of the Cluster of Excellence "Matters of Activity. Image Space Material" funded by the DFG under Germany's Excellence Strategy-EXC 2025-390648296.

**Institutional Review Board Statement:** The study was conducted according to the guidelines of the Declaration of Helsinki, following the data protection guidelines of University of Hamburg and the ethical guidelines of Deutsche Gesellschaft für Psychologie e.V. It was reviewed and acknowledged by the Local Ethics Commission of the Department of Informatics at University of Hamburg on 29 April 2021. The acknowledgement can be identified by the following name: CONDUCTORVR: COMPARING VR AND VIDEO-BASED EXERCISES IN OLDER ADULTS.

**Informed Consent Statement:** Informed consent was obtained from all subjects involved in the study. Written informed consent has been obtained from the participants to publish this paper.

**Data Availability Statement:** The data presented in this study are available on request from the corresponding author. The data are not publicly available due to possible identification of the subjects due to their age, gender, and place of residence.

**Acknowledgments:** We especially want to thank Hospital zum Heiligen Geist Hamburg for their support with participant acquisition and review of the application, and for giving us the possibility to conduct studies on their premises.

**Conflicts of Interest:** The authors declare no conflict of interest. The funders had no role in the design of the study; in the collection, analyses, or interpretation of data; in the writing of the manuscript; or in the decision to publish the results.

## Abbreviations

The following abbreviations are used in this manuscript:

| | |
|---|---|
| VR | Virtual Reality |
| VE | Virtual Environment |
| IVE | Immersive Virtual Environment |
| HMD | Head-Mounted Display |

## Notes

1   https://unity.com/accessed on 6 November 2021
2   https://beatsaber.com/ accessed on 6 November 2021
3   https://ohshapevr.com/ accessed on 6 November 2021
4   https://www.polar.com/products/accessories/oh1-optical-heart-rate-sensor accessed on 28 September 2021
5   https://jasp-stats.org/ accessed on 6 November 2021

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
