# Peer review of "Enabling Immersive Exercise Activities for Older Adults: A Comparison of Virtual Reality Exergames and Traditional Video Exercises"

_societies, doi:10.3390/soc11040134_

Round 1

Reviewer 1 Report

The paper present a comparative study between an exergame in VR and video-based fitness tutorials. While the topic has regain interest after the lockdowns imposed by the COVID-19 pandemic, it is not original. Thus there are already plenty of research about it. 

Authors should have referred more carefully on the theoretical background to for their hypothesis, which in the current version are poorly grounded. 

The exergame evaluated should be better described. Besides the description of the exercises there are no further details regarding the dynamics of the game, e.g, levels, challenges etc. 

Similarly, for the videos the exercises are only briefly described with no thorough information on how the contents were devised. Were any physiotherapists involved?  

Regarding the results, I noted that the values of the standard deviations of the NASA TLX are as high as the means. In this case data cannot be analyzed correctly without any further pre-processing. Authors should amend this. 

MINOR

The tool used to assess the heart rate is described in the procedure paragraph, it should be in a section dedicated to the equipment. 

Reviewer 2 Report

The manuscript deals with an experimental investigation focused on different performances in exercise activities for older adults. Traditional video exercises are compared with VR exergames. The manuscript is well structured and the topic is timely and innovative. I would like to mention the following four points which the authors should reflect and consider when preparing a revised version of the manuscript:

  • Could you make it clearer which of your three study hypotheses are related to which previous study results?

  • The discussion section hardly includes any references. Although the several study hypotheses could be confirmed, the chain of argumentation (How could these phenomena be explained based on previous literature?) remains rather vague.

  • What I miss in the discussion section is the aspect of immersion. In the VR condition, the participants apparently wore HMDs, which invites to a significantly different spatial presence than the screen-based video condition (see for e.g. https://doi.org/10.1007/s41064-020-00107-y). How could immersion have an impact on your results?

  • How did you consider aspects of spatial cognition in the immersive condition? Some studies point to different spatial perceptions and memory performances when using such a virtual reality environment (see, for e.g., effects on distance estimations: https://doi.org/10.3390/ijgi10030150). These effects specifically refer to the immersive VR condition and not to the screen condition. Those spatial aspects would be worth considering in the limitations (or background?) section.

Reviewer 3 Report

The paper presents the results of an original experiment that compared the virtual reality games against video game exercises. The experiment is explained in detailed and this is and the main strength of this paper. The results of the paper can be very useful to the scientific community and potentially attract citations. I enjoyed reading this paper, however, the paper has some serious issues that must be addressed before it can be published.

In the abstract remove the e.g and list what it was tested. (i.e., movement and heart rate) and i.e., enjoyment, attention allocation, and workload. Was there something else also tested? Kindly, please list everything that was tested.

Figure 1 is above the introduction. Move it in the body of the paper, e.g. the introduction section.

There is also an issue with the figures referencing in the text. The first reference of figure 1 in the text is after figure 3 in line 183. The same thing is happening with the rest of the figures as they are discussed in a different point. This needs to be fixed because it confuses the user.

Are the figures produced by you? If not, do you have the right to use them?

Provide a reference next to the first mention of  “exergame”  in order to introduce the term.

It is recommended to add some introductory text of section 2 at the end of section 1, after hypotheses. For example you can have 1-2 lines summarizing section 1 and state what comes next.  

In line 137, the authors have mentioned “as described in our previous paper [REMOVED FOR ANONYMITY].” but they mentioned MAESTRO VR Game! You should not mention “our previous paper”. Replace the text with “Researchers in [REFERENCE NUMBER]”  presented their results based on a VR game [REFERENCE NUMBER] or something similar. No journal will publish your paper with missing references as there is no way for us to validate your claims.

The authors may want to familiarize themselves with the concept of open research and the benefits that it provides to both the researchers as well as the scientific community. Open Research is a broader concept that goes beyond open access and MDPI is a leader in that field.  

The conclusion of the paper needs to extended, especially the first paragraph. Also summarize the limitations (from the previous section) in the conclusion and add a small future work section that will explain how the limitations can be addressed.  

The references of the paper must be revised as a number of references are not cited in the text. E.g 61 but more of them are NOT cited in the text. Also there are some jumps in the numbering in the text. Consider using automated tools to fix the references. Please note that this is a very major issue. Some journals have retracted published papers for missing references.

I would like to suggest the authors to provide a video abstract for this paper, outlining their results and the methodology.

Round 2

Reviewer 2 Report

The authors submitted a revised version of the manuscript. They take up the suggestions made by the reviewer and explained their changes in a response letter. The line of argumentation is clear. Against this background, I would like to propose thus current manuscript version for publication in this journal.

Reviewer 3 Report

Accept in its current form.